# Stereotactic Body Radiation Therapy (SBRT) for Oligorecurrent/Oligoprogressive Mediastinal and Hilar Lymph Node Metastasis: A Systematic Review

**DOI:** 10.3390/cancers14112680

**Published:** 2022-05-28

**Authors:** Salvatore Cozzi, Emanuele Alì, Lilia Bardoscia, Masoumeh Najafi, Andrea Botti, Gladys Blandino, Lucia Giaccherini, Maria Paola Ruggieri, Matteo Augugliaro, Federico Iori, Angela Sardaro, Cinzia Iotti, Patrizia Ciammella

**Affiliations:** 1Radiation Therapy Unit, Azienda USL-IRCCS di Reggio Emilia, 42123 Reggio Emilia, Italy; emanuele.ali@ausl.re.it (E.A.); gladys.blandino@ausl.re.it (G.B.); lucia.giaccherini@ausl.re.it (L.G.); mariapaola.ruggieri@ausl.re.it (M.P.R.); matteo.augugliaro@ausl.re.it (M.A.); federico.iori@ausl.re.it (F.I.); cinzia.iotti@ausl.re.it (C.I.); patrizia.ciammella@ausl.re.it (P.C.); 2Radiation Oncology Unit, S. Luca Hospital, Healthcare Company Tuscany Nord Ovest, 55100 Lucca, Italy; lilia.bardoscia@uslnordovest.toscana.it; 3Skull Base Research Center, Iran University of Medical Sciences, Tehran 1997667665, Iran; najafi.mas@iums.ac.ir; 4Medical Physics Unit, Azienda USL-IRCCS di Reggio Emilia, 42123 Reggio Emilia, Italy; andrea.botti@ausl.re.it; 5Interdisciplinary Department of Medicine, Section of Radiology and Radiation Oncology, University of Bari “Aldo Moro”, 70124 Bari, Italy; angela.sardaro@uniba.it

**Keywords:** oligorecurrent/oligoprogressive/oligometastatic disease, mediastinal and hilar lymph node metastases, SBRT, salvage treatment, stereotactic body radiation therapy, ablative radiotherapy, thorax dose-c

## Abstract

**Simple Summary:**

This paper is a review of the literature on oligorecurrent/oligoprogressive mediastinal and hilar lymph node metastasis treated with SBRT. The use of mediastinal SBRT had historically been not feasible in view of the expected toxicity due to the proximity of critical structures such as the airways and esophagus. Despite the heterogeneity and lack of some data in the studies analyzed, this literature review is the first published and can be a valid guide for the radiotherapist in the management of oligometastatic/oligoprogressive patients, with particular regard to the radiotherapy doses, dose constraints for organs at risk, and clinical outcomes.

**Abstract:**

Introduction: Mediastinal or hilar lymph node metastases are a challenging condition in patients affected by solid tumors. Stereotactic body radiation therapy (SBRT) could play a crucial role in the therapeutic management and in the so-called “no-fly zone”, delivering high doses of radiation in relatively few treatment fractions with excellent sparing of healthy surrounding tissues and low toxicity. The aim of this systematic review is to evaluate the feasibility and tolerability of SBRT in the treatment of mediastinal and hilar lesions with particular regard to the radiotherapy doses, dose constraints for organs at risk, and clinical outcomes. Materials and methods: Two blinded investigators performed a critical review of the Medline, Web of Knowledge, Google Scholar, Scopus, and Cochrane databases according to the Preferred Reporting Items for Systematic Reviews and Meta-Analyses statement (PRISMA), starting from a specific question: What is the clinical impact of SBRT for the treatment of oligorecurrent/oligoprogressive mediastinal and hilar metastasis? All retrospective and prospective clinical trials published in English up to February 2022 were analyzed. Results: A total of 552 articles were identified and 12 of them were selected with a total number of 478 patients treated with SBRT for mediastinal or hilar node recurrence. All the studies are retrospective, published between 2015 and 2021 with a median follow-up ranging from 12 to 42.2 months. Studies following SBRT for lung lesions or retreatments after thorax radiotherapy for stage III lung cancer were also included. The studies showed extensive heterogeneity in terms of patient and treatment characteristics. Non-small cell lung cancer was the most frequently reported histology. Different dose schemes were used, with a higher prevalence of 4–8 Gy in 5 or 6 fractions, but dose escalation was also used up to 52 Gy in 4 fractions with dose constraints mainly derived from RTOG 0813 trial. The radiotherapy technique most frequently used was volumetric modulated arc therapy (VMAT) with a median PTV volume ranging from 7 to 25.7 cc. The clinical outcome seems to be very encouraging with 1-year local control (LC), overall survival (OS) and progression-free survival (PFS) rates ranging from 84 to 94%, 53 to 88% and 23 to 53.9%, respectively. Half of the studies did not report toxicity greater than G3 and only five cases of fatal toxicity were reported. CONCLUSIONS: From the present review, it is not possible to draw definitive conclusions because of the heterogeneity of the studies analyzed. However, SBRT appears to be a safe and effective option in the treatment of mediastinal and hilar lymph node recurrence, with a good toxicity profile. Its use in clinical practice is still limited, and there is extensive heterogeneity in patient selection and fractionation schedules. Good performance status, small PTV volume, absence of previous thoracic irradiation, and administration of a high biologically effective dose (BED) seem to be factors that correlate with greater local control and better survival rates. In the presence of symptoms related to the thoracic lymph nodes, SBRT determines a rapid control that lasts over time. We look forward to the prospective studies that are underway for definitive conclusions.

## 1. Introduction

Isolated lymph node recurrence presents a challenge for physicians who are constantly seeking to develop a local, ablative, and targeted therapeutic approach in order to postpone systemic treatments. In this context, stereotactic body radiation therapy (SBRT), also known as stereotactic ablative radiotherapy (SABR), has a key role, as it allows delivery of high doses of radiation in relatively few treatment fractions with excellent sparing of healthy surrounding tissues and low toxicity.

Since its development in the 1990s, SBRT has emerged as one of the most significant advances in the modern era. By utilizing accurate target delineation, motion management, conformal treatment planning, and daily image guidance, SBRT is able to deliver high doses in few fractions and provide a steep dose fall-off outside the target with low toxicity [1,2,3,4]. The pioneering prospective dose-finding SBRT study was conducted at Indiana University and determined maximal tolerable doses in 47 medically inoperable patients to be 3 × 20 Gy and 3 × 22 Gy for T1 and T2 lesions, respectively [5]. Subsequently, Timmerman et al. published the results of 70 patients with T1-2N0 inoperable tumors who were treated with 60 and 66 Gy in three fractions. Despite an excellent 2-year LC of 95%, toxicity was unacceptably high [6]. To date, there have been three randomized control trials comparing surgery vs. SBRT in operable patients (ROSEL, STARS, RTOG 1021/ACOSOG Z4099), all of which have closed due to poor accrual. Despite this, a pooled analysis of patients from the STARS and ROSEL trials offers potential insight. In this analysis, a total of 58 patients were analyzed with an estimated 3-year OS of 95% for SABR vs. 79% for lobectomy, with a median follow-up of 40.2 months for the SBRT group and 35.4 months for the surgery group [7,8].

Moreover, several experiences regarding the treatment of various sites of macroscopic lymphadenopathies have been reported (i.e., pelvic or abdominal lymph nodes), with promising results in terms of local control, symptom control, and systemic therapy-free survival [9,10,11,12,13,14]. Furthermore, modern imaging techniques, in particular nuclear medicine, allow the early detection of isolated disease progressions and lead to the implementation of SBRT in different clinical scenarios.

The concept of oligometastatic disease was first introduced by Hellman and Weichselbaum about 20 years ago [15] and currently describes an intermediate state between localized tumor and diffuse metastatic disease, characterized by a limited number of metastases (usually 1–5) in a few organs (usually 1–3) and a more indolent behavior [16]. The concepts of “oligoprogression”, “oligorecurrence” and “oligopersistence” can be considered sub-categories of the broader definition of oligometastatic disease and describe a pathological state characterized by better prognosis in which outcomes can be improved with the use of local treatment [17].

Mediastinal or hilar lymph node recurrences are mostly due to non-small cell lung cancer (NSCLC) dissemination, considering that approximately 15–20% of patients with stage I disease [18] and up to 50% of patients with stage III disease will develop locoregional relapse [19]. However, other solid tumors, such as colorectal cancer, renal cancer and breast cancer may also involve the thoracic lymph nodes. Historically, therapeutic approaches in this setting involve the use of surgery, chemotherapy, or the combination of chemotherapy with conventional fractionation radiotherapy. The use of SBRT has only recently developed, as the thoracic district, and in particular, the anatomical structures it contains, such as the great vessels, the bronchi, and the heart, have always represented an obstacle to the use of high doses of radiotherapy, above all for retreatment. The objective of the present systematic review is to evaluate the clinical outcomes, in terms of overall survival (OS), local control (LC), and toxicity of the use of SBRT in mediastinal and hilar lesion treatment with or without previous thoracic irradiation, with a particular interest in the radiotherapy doses used and dose constraints for organs at risk.

## 2. Materials and Methods

### 2.1. Search Strategy

The present systematic review followed the guidelines for the Preferred Reporting Items for Systematic Review and Meta-Analysis (PRISMA) revised in 2015 [20] and was generated by the following question: What is the clinical impact of SBRT for the treatment of oligorecurrent/oligoprogressive mediastinal and hilar metastasis?

Databases including Medline, Web of Knowledge, Google Scholar, Scopus, and Cochrane were searched by two blinded investigators for all eligible studies based on the considered keywords, including “mediastinal and hilar lymph nodes”, or “treatment of mediastinal and hilar lymph nodes”, or “stereotactic body radiation therapy for mediastinal and hilar lymph nodes”, or “SBRT for mediastinal and hilar lymph nodes”, or “SBRT for thoracic lymph node metastasis”, or “thoracic oligometastasis treatment”, or “mediastinal and hilar lymph node oligometastasis treatment”, or “mediastinal and hilar lymph node recurrence”, or “SABR”, or “mediastinal and hilar lymph node oligoprogression” and “mediastinal and hilar lymph node oligorecurrence”.

The research involved an analysis of all studies published up to February 2022.

### 2.2. Selection Criteria

The inclusion criteria were as follows: (1) retrospective and prospective clinical trials; (2) studies published in the English language; (3) studies after SBRT on lung lesions or retreatments after thorax radiotherapy for stage III pulmonary neoplasms were also included. Exclusion criteria were: (1) lack of access to the full text of the manuscript; (2) studies with unclear or irreproducible results (i.e., lack of clear outcomes or presence of errors in methodology and/or analyses); (3) case reports; (4) case series; and (5) review papers.

### 2.3. Data Extraction

The data collection was independently performed by two unblinded reviewers on structured collection forms. We resolved disagreements by consensus or by involving a third person.

The extracted data consisted of the author, year of publication, study design, sample size, primary tumor, stage at the time of treatment (oligoprogressive/oligometastatic or oligorecurrent), follow-up, range of SBRT dose, dose per fraction, number of fractions, prescription isodose, PTV volume, previous thoracic irradiation, overall survival (OS), local control (OS), progression-free survival (PFS), and toxicity.

## 3. Results

According to the purpose of the study, in the initial search with keywords, 552 articles were identified. In the first step, the 552 studies were reviewed by title and abstract. Then, studies that did not meet the inclusion criteria were excluded. In the second step, the full text of 33 studies was reviewed. Finally, 12 studies were selected (Figure 1) with a total number of 478 patients. Figure 1 shows the flowchart for screening the eligible studies. Table 1 and Table 2 summarize patient characteristics and treatment outcomes, respectively. Franceschini and Kowalchuk [21,22] both published two papers on the use of SBRT in the treatment of mediastinal lymph node oligometastases. In both cases, the authors did not clearly specify whether the most recent article provided an update of the patient follow-up of the first manuscript, so after internal discussion, it was decided to consider the articles separately. All 12 selected studies are retrospective and were published between 2015 and 2021. Median follow-up ranged from 12 to 42.2 months. In many cases, the radiotherapy technique was described, but the most widely used was volumetric modulated arc therapy (VMAT) and more rarely CyberKnife (CK). The planning target volume (PTV) varied widely in the different studies, with a median PTV ranging from 7 to 25.7 cc. Unfortunately, the number of treated lesions is not reported in most of the studies. This data can be extracted in just five studies [21,23,24,25,26]. One, two and three lymph nodes were irradiated in 188, 37 and 11 patients, respectively. In one study, irradiation with four lymph nodes was described. The total number of lymph nodes undergoing SBRT was 299.

### 3.1. Age, Sex and Performance Status

Across all studies, except for Meng et al. [23] and Kowalchuk et al. [22], the youngest and oldest patients were 29 and 89 years old, respectively. Age data are not present in the study by Kowalchuk et al., while only a subdivision among patients aged under 60 (eight patients) or over 60 (nine patients) is reported in the study by Meng et al. Sex is equally distributed between females and males, 211/420 (50.2%) and 209/420 (49.8%), respectively. However, these data are missing in one study [22].

Two scales were used to evaluate the performance status: Karnofsky performance score (KPS) [25,30,32] and Eastern Cooperative Oncology Group (ECOG) [21,24,26,27,28,29], while in 3 studies it is not described [22,23,31]. Most patients had an ECOG between 0 and 1 and a KPS > 70.

### 3.2. Doses and Fractionation

The management and doses in this subset of patients still remain unclear, as in the past there was little evidence on the efficacy and safety of SBRT on hilar and mediastinal lymph node lesions, and this explains the wide variation in dose and fractionation used in the different studies. Fractionation into five or six fractions is the most commonly used in the majority of the studies, with a dose per fraction ranging from 4 to 10 Gy. More accelerated fractionations, such as 48–52 Gy in four fractions, are also used in some cases [29], as well as schedules in eight sessions for a total dose of 48 or 56 Gy. Franceschini et al. [21] reported a median biologically effective dose (BED) delivered of 75 Gy in a series of 72 patients, while a higher median BED was described by Meng et al. [13], Shahi et al. [26] and Wang [25], at 83 Gy, 116.7, and >100 Gy, respectively. In most reports, it was not possible to discriminate the doses used in the retreatments compared to the doses of patients who had not received previous radiotherapy.

### 3.3. Primary Tumor

The most frequent primary tumor that gave rise to mediastinal or hilar lymph node metastases treated with SBRT is lung cancer, with 320 out of 478 patients (66.9%), in which non-small cell lung cancer (NSCLC) is the most widely represented histological variant (over 90% of cases). All studies reported cases of lung cancer, while six had only NSCLC histology as inclusion criteria [22,23,27,29,30,32]. The other primary tumors present, in order from the most to the least frequent, are kidney cancer, 35 pts (7.3%), breast cancer, 30 (6.3%), colorectal cancer, 21 (4.4%), head and neck cancers, 13 (2.7%), upper gastrointestinal cancers, 13 (2.7%), gynecological cancer, 9 (1.9%), prostate cancer, 3 (0.6%), and hepatocarcinoma, 3 (0.6%).

In six studies, 194 patients presented with oligorecurrent disease [21,23,24,27,29,32], while five studies [22,25,26,30,31] included patients with both recurrent and oligoprogressive disease, for a total of 277 patients. Finally, Yeung et al. included only oligometastatic patients [29].

### 3.4. Previous Treatments

Considering the oligometastatic, oligorecurrent or oligoprogressive nature of the disease, 182 patients underwent previous or concurrent systemic treatments to SBRT. In 198 cases, no chemotherapy was administrated previously to or in combination with radiotherapy. In three studies [22,23,28], no information on systemic treatments was reported.

With regard to radiotherapy, 145 patients (30.3%) had received prior thoracic radiotherapy treatment, generally characterized by conventional doses (range 57–70 Gy) of radiotherapy for the treatment of locally advanced stage III non-small-cell lung cancer; therefore, in these cases the subsequent stereotactic treatment on the lymph node progression sites configured as a reirradiation or previous SBRT for stage I-II lung cancer (in one to four fractions). Conversely, 201 (42%) cases had not received prior thoracic radiotherapy. However, these data are not reported in five studies [22,24,27,28,29].

### 3.5. Dose Constraints

Not all articles detail the dose constraints used for organs at risk. Shahi et al. and Wang et al. stated that they used the Radiation Therapy Oncology Group (RTOG) recommendations 0236 and 0813 [33,34], Horne et al. used the 2017 edition of the National Comprehensive Cancer Network (NCCN), while Franceschini et al. [24] reported the internal constraints of their institution. A summary of the constraints mainly used in five fractions is given in Table 3.

It should be emphasized that the constraints proposed by the NCCN are also derived from the RTOG 0813 studies. Few studies reported the radiotherapy technique and the prescription of the isodose at the PTV.

### 3.6. Clinical Outcomes (LC, OS, PFS)

Median follow-up ranged from 12 to 42.2 months, and the one-year local control rate obtained ranged from 84% to 94%. Franceschini et al. even reported 47/76 cases of complete response to treatment [21], while Wang reported 47 out of 85 patients and Jereczek-Fossa 11 out of 42 patients [31]. Furthermore, local control rates close to 70% at 2 and 3 years are reported in three studies [23,24,32]. Only two studies give the five-year LC: 58% and 77% reported by Manabe [29] and Wang [25], respectively.

Regarding the overall survival data, even though they concern pathologies in oligoprogression/oligorecurrence, they are encouraging: one-year OS ranged from 53% to 88%. Studies reporting two-year data confirm an OS greater than 60%, while at three years it is around 40% [23,24]. Menabe et al. and Wang are still the only authors who report survival at 5 years, which drops to 14% and 21%, respectively.

Regarding progression-free survival, the data change considerably depending on the study and the primary pathology. The 1y-PFS ranged from 23% to 53.9%, while the 2y-PFS dropped below 20%.

Meng et al. [23] found an enormous benefit in symptom control only 6 days after the end of RT and lasting in the follow-up.

Shahi et al. [26] found that two out of three patients (66.8%) at 1 year, and 42.9% of patients at 2 years, did not require a change in their strategy.

### 3.7. Toxicity

In many studies, the toxicity scale used was not reported, therefore the collection of results could be biased by this omission.

Only five cases of fatal toxicity (1.3% of the total number of patients) were described by Manabe et al. [29], in which a patient died of pneumonitis, and Wang et al. [25] due to a tracheoesophageal or esophageal-mediastinal fistula as a consequence of reirradiation. Meng et al. [23] described a G5 toxicity in reirradiation group patients after radio-chemotherapy treatment for lung cancer. No other G5 toxicity was reported. Two studies reported two patients (0.4%) experiencing G4 myocardial toxicity [21,24], and one patient with G4 hemoptysis. A total of three patients (0.6%) experienced G4 toxicity [30]. However, half of the studies did not report toxicity greater than G3. In cases < G2 toxicity, characterized by pneumonitis and esophagitis, was more frequent (around 50%).

### 3.8. Prognostic Factors

Statistical analysis for the determination of prognostic factors that may influence outcomes and toxicity was performed in 9 of 12 studies [21,22,23,24,25,29,30,31,32]. The most important prognostic factor related to overall survival was good performance status [21,24,25,29]. Other factors positively related to OS were: small PTV [25,29,32], previous or concurrent chemotherapy [21,29,32], long interval time between primary treatment and salvage SBRT >12 months [23,25] or >15 months [29], previous surgical treatment rather than previous SBRT for lung lesions [29], over 65 years of age [24], and the absence of symptoms [25]. Instead, features that had a negative impact on OS were primary colorectal [21] or mammary cancer [24] and large PTV.

Local control was instead positively influenced by performance status [21,25,29,30], female sex [30], previous or concomitant chemotherapy [24,29,32] and small PTV [30,31,32]. Another important factor for the local control of the disease was the total dose, the BED, and the dose per fraction. The following were identified as the cut-off: total dose >60 Gy [29] or BED > 60 Gy [31], or >75 Gy [21], or >100 Gy [32], a dose per fraction > 8 Gy [31].

Regarding toxicity, previous radiotherapy and therefore reirradiation was the only factor related to high toxicity [23,25,31,32].

## 4. Discussion

Historically, stereotactic body radiation therapy has been used in patients with peripherally located lung tumors, since high-dose RT of centrally located tumors close to critical organs (such as the esophagus, bronchial tree, heart, and great vessel) was thought to possibly cause severe toxicities [35]. As a consequence, tumors within a 2 cm radius of the proximal bronchial tree were described as the no-fly zone (NFZ) and for a long time were excluded from high-dose ablative treatments. In 2019, the RTOG 0813 study, which investigated the use of five-fraction SBRT in central and ultra-center lesions, provided robust data on the safety and efficacy of SBRT, concluding that it is well tolerated and is associated with relatively low rates of serious treatment-related disease toxicity in this setting [34,36]. There is less certainty regarding the treatment of mediastinal and hilar lymph nodes with the SBRT technique. The treatment of this patient setting has always represented a challenge for radiotherapy, and the fear of possible side effects has led to the use of a risk-adaptive fractionation strategy, with the use of palliative doses, such as 30 Gy in 10 fractions, resulting in minimal toxicity and unsatisfactory local control. Consequently, the ideal treatment for this subset of patients remains unclear, and systemic therapies (chemo-, endocrine- and biological therapy) aimed at prolonging survival and preventing or controlling symptoms are still the standard treatment. The introduction of the concept of oligometastatic disease has revolutionized the management of stage IV disease. The use of local ablative therapy, such as surgery, and above all high-dose radiotherapy has been shown to increase local control of disease, and in some cases even OS, sometimes even delaying the initiation of systemic therapy [35,37,38,39].

In 2015, Meng et al. [23] first investigated the efficacy of SBRT in thoracic lymph node disease after SBRT for lung tumors. The authors identified that a time between surgery and SBRT of fewer than 15.5 months was a negative prognostic factor of OS, a sign of a potentially more aggressive disease. They also found an advantage, although not statistically significant, of administering chemotherapeutic agents in addition to SBRT alone. Other prognostic factors were PTV and performance status. It should be noted that Meng et al. [23] found an enormous benefit in symptom control only 6 days after the end of RT and lasting in the follow-up. Despite the low toxicity rates, they concluded that extreme caution should be exercised in the use of SBRT in patients who have previously received RT, particularly in station 7 irradiation, in consideration of the possible large overlap of the PTV with the airways.

The following year, the Cleveland group [27] questioned the role of SBRT in lymph node recurrence from NSCLC. Their suggestion was to reserve SBRT only for highly selective patients and to treat patients in intermediate condition with conventional treatment in 15 sessions and patients in good condition with 30 sessions, possibly in association with chemotherapy, as suggested by the Radiation Therapy Oncology Group [40,41]. The milestone study in this context, with the largest case series (85 patients) and largest follow-up (median 42.2 months), was conducted by Wong et al. [24]. The authors confirmed in these papers that worse OS is associated with short intervals since previous RT, poor patient performance status, and large PTV, as suggested by Meng et al. However, they draw attention to the fact that by using a dose with BED10 >100 Gy, local control can be guaranteed even on large PTV volumes, while not influencing OS. Indeed, they report a 5 y actual LC rate of 77% higher than reported rates for conventional RT as well as for OS [42,43,44]. Similar results emerged in the Kowalchuck study [22], in which in addition to patients with lymph node metastases, 42 patients in locally advanced stage (cT3-4 cN1-3) treated with SBRT were also considered. The authors suggest considering SBRT also in stage III NCLC, as their study demonstrated comparable local control rates. We believe, instead, that it is appropriate to reserve this treatment for highly selective patients, possibly elderly ones who are not susceptible to systemic treatment. Mediastinal lymph node recurrences from NSCLC, and even more stage III tumors, should be treated as locally advanced disease, even if the recurrence appears after years, as real-life scientific evidence has shown a long-term benefit of concurrent chemoradiotherapy treatments, possibly associated with maintenance durvalumab [44,45].

This study also showed the need to be cautious in the reirradiation of stage 7 lymph nodes due to the risk of lethal toxicity (three patients) in consideration of the risk of necrosis or fistula of the airways, as also described in a case report [46].

Franceschini et al. [24] found a benefit of SBRT in elderly patients, probably because they are referred early to radiotherapy treatment due to co-pathology and performance status or due to the presence of more indolent disease. Surprisingly, the study data also demonstrated a worse prognosis for breast metastases, which was unforeseen by the authors themselves since breast oligometastatic disease seems to benefit greatly from SBRT treatment [47]. This can be explained by the presence of significant biases in the study, the first being the small sample size, which did not enable a correct statistical analysis to be performed. Secondly, there could have been incorrect staging of the patients with a multimetastatic and non-oligometastatic stage disease, and the impact of Luminal A/B or triple-negative or HER2-like subcategories was not analyzed. In fact, these data are refuted in the second analysis of the Milan institute [21], in which it emerged that the histology that least benefits from SBRT in oligometastatic sites is colorectal adenocarcinoma, confirming the well-known radio-resistance of this type of tumor.

In these latter studies, two issues are touched upon which, in the opinion of the authors of this present study, are of considerable importance. The first is the need to discuss in a multidisciplinary context the treatment that could best be of benefit to oligometastatic/oligorecurrent/oligoprogressive patients, which could help to correctly define the risks and benefits of each single treatment (surgical vs. chemotherapy vs. radiotherapy). In the second instance, there is the need to define the dose that allows the greatest local control and the least possible toxic effect. A dose with BED greater than 75 Gy correlates with higher local control rates in the Franceschini study. However, the one-year local control rates of Meng et al. [23] are significantly higher compared to Franceschini (100% vs. 86.6%) because the equivalent dose administered was greater than 83 Gy. Several studies [48,49] regarding the treatment of central and ultra-central lesions found that a BED >100 Gy significantly improved both local control and overall survival, while there was no advantage in LC with a dose higher than 120 Gy, as suggested by Jereczek-Fossa [31].

As regards the data relating to progression-free survival, one-year PFS varied from 3% to 53.9%, while at 2 years it was less than 20%. Considering that these are stage IV patients, who have often come to SBRT after multiple lines of treatment, the data are encouraging. There is no doubt that these data raise a significant question, namely the correct selection of patients. It is conceivable that patients were included who would not have benefited from the SBRT treatment from the beginning. Not only that, but in many cases purely palliative treatments have been included.

Finally, Shahi et al. [26] analyzed a topic of extreme interest and which actively stimulates current research, namely whether SBRT can delay the initiation of systemic treatments. There may be several potential benefits of delaying changing systemic therapy: (1) prolonged breaks from systemic treatment may allow for quality of life preservation; (2) targeting progressive drug-resistant clones may allow current lines of systemic treatment to continue and prevent or delay the need to start subsequent (and potentially more toxic) lines of treatment; and (3) the use of locally ablative therapies at the time of disease progression may be more cost-effective than the traditional strategy of changing to next-line therapy [50].

The authors [26] reported that although distant progression was common, more than half (54%) of patients received further SBRT or SRS after initial mediastinal SBRT, indicating that the salvage of distant failures was feasible and may have additionally delayed systemic treatment. In fact, they found that two out of three patients (66.8%) at 1 year, and 42.9% of patients at 2 years, did not require a change in their strategy even if it did not reach statistical significance, but a trend is clearly visible. Furthermore, subsequent disease progressions are often characterized by pictures of oligoprogression, which in turn may be susceptible to further stereotactic treatment.

Currently, the interaction between SBRT and the new drugs available is of great interest; in fact, the advent of immune checkpoint inhibitors (ICIs) has dramatically changed the landscape of cancer care, since immunotherapeutic strategies are emerging as potentially curative systemic therapy for several tumors, especially for tumor types traditionally known to have poor outcomes. Treatment with a fully human anti-cytotoxic T-lymphocyte–associated protein 4 (CTLA-4) antibody has been associated with long-lasting responses in several hematologic malignancies [51] and a high proportion of durable, complete responses in patients with advanced metastatic melanoma [52]. Anti-programmed cell death 1 (anti-PD-1) or programmed death-ligand 1 (PD-L1) blocking antibodies have shown objective responses in a variety of solid tumors, including melanoma, lung cancer, prostate cancer, breast cancer, ovarian cancer, head and neck cancer, and a subset of colorectal cancers [53,54,55,56,57,58]. Immune checkpoint blockers have been demonstrated to enhance durable disease responses in both early and advanced tumor settings, alone or in combined strategies, as well as improving or retaining the patient’s quality of life.

Some studies have reported that concurrent and nonconcurrent treatment with SBRT and checkpoint inhibitors achieve better outcomes with no increased toxicity [59,60,61,62,63,64,65,66,67,68]. However, others warn of possible immune-related adverse events and a synergistic effect of radiotherapy and immunotherapy on toxicities [69]. Moreover, the addition of SBRT could trigger a reactivation of the immune response in patients who are no longer responsive to immunotherapy, triggering an ex novo immune response [70].

The present review of the literature, despite focusing on retrospective studies, gives new perspectives on a strategy that is still little explored albeit with numerous limitations. The main limitations of the study relate to the retrospective nature of the studies analyzed, and the absence of prospective studies makes the consensus less solid. The heterogeneity of the collected studies, in terms of patient characteristics, doses, pathologies, and stage of disease, did not enable the carrying out of statistical analysis with strong bases; therefore, the comparison of studies, which is sometimes difficult, has a purely descriptive nature. Moreover, from the analysis of the texts, it was not possible to discriminate the doses used in the retreatments compared to the doses of patients who had not received previous radiotherapy. Prospective studies are currently ongoing and are registered on clinicaltrial.gov (NCT02019576, NCT02756793, NCT03256981, and NCT03644303)(Accessed on 5 February 2022). As soon as the results are available, we will be able to answer questions that are still open.

## 5. Conclusions

From the present review, it is not possible to draw definitive conclusions because of the heterogeneity of the studies analyzed, and, furthermore, it is not conceivable to perform a meta-analysis due to the limited number of studies present. However, it is possible to deduce some important considerations.

Ablative SBRT for oligoprogressive/oligorecurrent/oligometastatic mediastinal and hilar lymph nodes is both effective in terms of local control and safety, although the analysis is derived from small retrospective studies with relatively short follow-up, while data about PFS or OS could not be derived from these data due to the absence of a control group. Safety is also confirmed by studies that have investigated SBRT for ultra-central thoracic lesions, in which the risk of fatal toxicity, albeit low, should not be underestimated. Moreover, also as a general and intuitive consideration, very close attention should be paid to the reirradiation of lymph node metastasis, because there may be a high PTV overlapping with the airways with a consequent risk of high toxicity. Therefore, a multidisciplinary discussion is mandatory for the evaluation of the real risks and benefits on a case-by-case basis. The authors suggest that for lymph node NSCLC oligorecurrence, SBRT should be reserved for selected cases, while concurrent conventional chemoradiation is preferable, if not previously irradiated. Good performance status, small PTV volume, the absence of previous thoracic irradiation, and the administration of high BED dose seem to be factors that correlate with greater local control and better survival rate. In the presence of symptoms related to the thoracic lymph nodes, SBRT determines a rapid control that lasts over time. Moreover, also in this setting, SBRT could delay the initiation of systemic treatments.

Finally, the authors of this study recommend an RT dose delivered in four to six fractions with a BED greater than 80–100 Gy. Prospective studies are necessary to confirm the current evidence and to answer the questions still open.

## Figures and Tables

**Figure 1 cancers-14-02680-f001:**
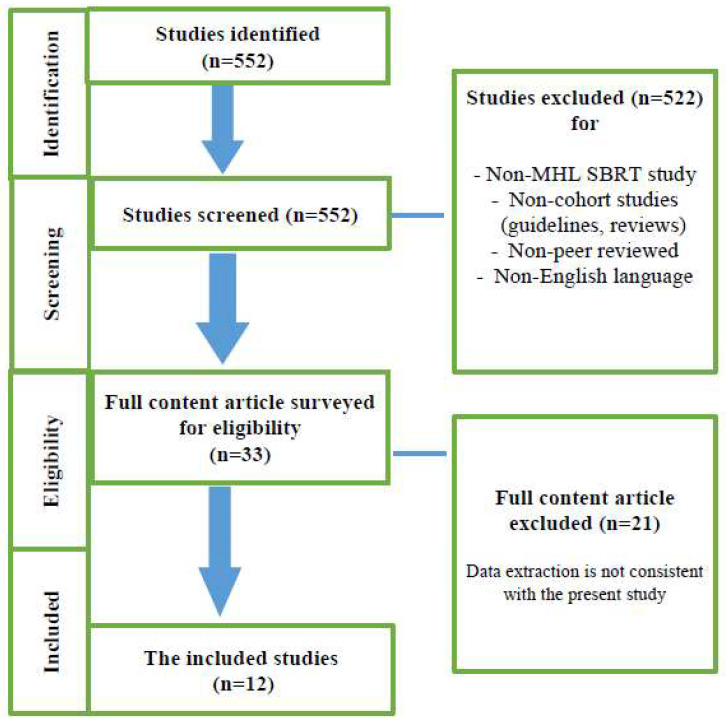
The flowchart for screening the eligible studies. Abbreviations: MHL: mediastinal-hilar lymph node; SBRT: Stereotactic body radiation therapy.

**Table 1 cancers-14-02680-t001:** Summary of patient characteristics.

	Meng et al. [23]	Ward et al. [27]	Franceschini et al. [24]	Wang et al. [25]	Yeung et al. [28]	Manabe et al. [29]	Horne et al. [30]	Jereczeck-Fossa et al. [31]	Franceschini et al. [21]	Kowalchuk et al. [32]	Shahi et al. [26]	Kowalchuk et al. [22]
N.	33	10	29	85	7	14	40	42	76	32	52	58
Age	8 pt < 60y 9 pt ≥ 60y	M. 77 y (56–87)	M. 67 y (24–84)	M. 59 y (32–89)	M. 65 y (31, 88)	M. 77 y (29–87)	M.70 y (47–95)	M. 62.3 y (43–84)	M. 62.6 y (47–72)	M. 76.13 y	M. 65 y (43–85)	NA
Sex Male Female	23 10	4 6	16 13	54 31	4 3	11 3	16 24	17 25	37 39	14 18	13 39	NA
Median KPS ECOG 0 ECOG 1 ECOG 2 ECOG 3	NA	18 10 1	18 10 4 0	80 (60–90)	ECOG ≤ 1 17 ECOG ≥ 2 1	2 5 6 1	80 (70–100)	NA	43 31 2 0	KPS = 70 23% pt KPS > 70 77% pt	29 23	NA
Tumor	Lung	Lung	Lung 12 Breast 4 CL 6 Other 7	Lung 53 Esoph. 7 Breast 7 HCC: 3 HN: 7 Kidney 3 Bladder 2 GYN 2 CL 1	NA	Lung	Lung	Lung 25 Breast 5 GYN 7 HN 3 Anus 1 CL 1	Lung 35 CL 10 UGI 6 Breast 10 Kidney 4 Other 11	Lung	Lung 8 Kidney 28 Breast 4 HN 3 CL 3 Prost 3 Other 3	Lung
Previous mediastinal RT	Yes 19 No 14	NA	NA	Yes 29 No 56	NA	NA	Yes 20 No 20	Yes 11 No 31	Yes 30 No 46	Yes 24 No 8	Yes 12 No 40	NA
Chemotherapy	NA	Yes 4 No 6	Yes 9 No 20	Yes 49 No 36	NA	Yes 0 No 14	Yes 34 No 6	Yes 30 No 12	Yes 21 No 55	Yes 0 No 32	Yes 35 No 17	NA

Abbreviations: N: number of patients; y: year; NA: Not available; M.: Median age; ECOG: Eastern Cooperative Oncology. Group; KPS: Karnofsky performance score; CL: colorectal cancer, HN: Head and neck cancer, Prost: Prostate, GYN: Gynecological cancer, UGI: upper gastrointestinal cancer.

**Table 2 cancers-14-02680-t002:** Studies selected for systematic review: treatment characteristics and clinical outcomes.

Author	Year	State of Disease	Median FUP (Months)	Dose Range (Gy)/Daily Dose (Gy)/N. Fractions	PTV Volume CC (Range)	Local Control	OS	PFS	Toxicity
Meng et al. (Retrospective) [23]	2015	OR	20.9	24–60/3–18/3–15	17.89 (4–145)	3 y: 86%	3 y: 40.7%	NA	Any: G1–G2: 6 pts G3: 3 pts
Ward et al. (Retrospective) [27]	2016	OR	11.3	17–45/2.5–8.5/2–20	NA	1 y:84.4%	1 y: 53%	1 y: 33%	No tox > G3
Franceschini et al. (Retrospective) [24]	2016	OR	12	30–60/6–7.5/5–8	NA	14 CR	1 y: 76	1y: 28	Cardiac: G4: 1 pts
Wang et al. (Retrospective) [25]	2016	OM/OP	42.2	45–60/5–18/3–10	15.3	1 y: 97% 5 y: 77%	1 y: 78.2% 2 y: 43.6% 5 y: 21.3%	NA	Lung: G3: 4 pts G5: 3 pts
Yeung et al. (Retrospective) [28]	2017	OM	33.6	31–60/5–8/4–10	34.8 cc (6.5–162.2)	1 y: 94%; 2 y: 47%	1 y: 89% 2 y: 74%	1 y: 39%; 2 y: 17%	No ≥ G3
Manabe et al. (Retrospective) [29]	2018	OR	11	48–52/10.5–12.5/4	13 (5.9–23)	5 y: 58%	5 y: 14%	5 y: 21%	Lung: G2: 6 pts G3: 5 pts G5: 1 pts Esophagus: G3: 1pts
Horne et al. (Retrospective) [30]	2018	OM/OR	16.4	35–48/7–12/4–5	7.25 (0.7–88.3)	1 y: 87.7%	1 y: 69.2	1 y:53.9	Lung: G4: 1 pts Esophagus: G1:18 pts G2:1 pts
Jereczeck-Fossa et. al (Retrospective) [31]	2018	OM/OP	16	21–50/5–12/3–5	NA	1 y: 66.3%	1 y: 88.3%	NA	No ≥ G2
Franceschini et al. (Retrospective) [21]	2019	OR	23.16	25–60/5–10/5	NA	2 y: 68%	3 y: 41.6	3 y:16.4%	Lung: G2: 3 pts Cardiac: G4: 1pts
Kowalchuk et al. (Retrospective) [32]	2020	OR	23	15–50/6–12/4	15.5 (1.28–269.6)	64%	23.51 mo	15.34 mo	No tox > G3
Shahi et. al (Retrospective) [26]	2020	OM/OP	20	30–50/6–10/5	NA	2 y: 91%	1 y: 84.2%; 2 y: 67.8%	1 y: 23.6%; 2 y: 11.6%	Lung: G3: 5 pts Esophagus: G3: 1 pts G4: 1 pts
Kowalchuk et al. (Retrospective) [22]	2021	OM/OP	17.5	5–60/5–15/1–5	25.7 (1.28–69.6)	2 y: 77%	2 y: 65%	2 y: 48%	NO > G3

Abbreviations: FUP: follow-up; PTV: Planning Target Volume; OS: Overall Survival; PFS: Progression-free survival; OR: oligorecurrent; OM: oligometastatic; OP: oligoprogressive, y: year; mo: months; Tox: toxicity, PTS: patients, NA: not available; G: Toxicity Grade.

**Table 3 cancers-14-02680-t003:** Summary of the constraints used in the articles. For simplicity, only the constraints in 5 fractions have been reported.

*OARs*	*RTOG 0813* Used by [26,27]	*Franceschini* et al. Used by [24]	*RTOG 0236* Used by [26,27]	*NCCN V. 2017* Used by [30]
*Combined Lungs*	V_12.5Gy_ ≤ 1500 cm^3^ V_13.5Gy_ ≤ 1000 cm^3^	V_5Gy_ ≤ 30% V_10Gy_ ≤ 17% V_20Gy_ ≤ 12% V_30Gy_ ≤ 7%	V_20Gy_ ≤ 15%	NA
*Spinal Cord*	D_max_ < 30 Gy (6 Gy/fx) V_22.5Gy_ < 0.25 cm^3^ (4.5 Gy/fx) V_13.5Gy_ < 0.5 cm^3^ (2.7 Gy/fx)	V_22.5Gy_ < 0.25 cm^3^ V_13.5Gy_ < 0.5 cm^3^ D_max_ < 30 Gy	18 Gy (6 Gy/fx)	D_max_ < 30 Gy (6 Gy/fx)
*Esophagus*	D_max_ < 105% V_27.5Gy_ < 0.5 cm^3^ (5.5 Gy/fx)	D_max_ < 105% D_1 cm3_ < 30 Gy D_5 cm3_ < 27.5 Gy	27 Gy (9 Gy/fx)	D_max_ < 105%
*Stomach*	NA	D_max_ < 105% D_1 cm3_ < 30 Gy D_5 cm3_ < 27.5 Gy	NA	NA
*Heart*	D_max_ < 105% V_32Gy_ < 1.5 cm^3^ (6.4 Gy/fx)	D_max_ < 105% D_1 cm3_ < 40 Gy D_5 cm3_ < 20 Gy D_15 cm3_ < 32 Gy	30 Gy (10 Gy/fx)	D_max_ < 105%
*Large Vessels*	D_max_ < 105% V_47Gy_ < 1 cm^3^ (9.4 Gy/fx)	D_max_ < 105% D_1 cm3_ ≤ 40 Gy D_10 cm3_ < 47 Gy	NA	D_max_ < 105%
*Main bronchus/trachea*	D_max_ < 105% V_18 Gy_ < 0.4 cm^3^ (3.6 Gy/fx)	D_max_ < 105% D_4 cm3_ < 40 Gy V_35Gy_ ≤ 1 cm^3^	30 Gy (10 Gy/fx)	D_max_ < 105%
*Brachial Plexus*	D_max_ < 32 Gy (6.4 Gy/fx) V_30Gy_ < 0.3 cm^3^	NA	24 Gy (8 Gy/fx)	D_max_ < 32 Gy (6.4 Gy/fx)
*Skin*	D_max_ < 32 Gy (6.4 Gy/fx) V_10Gy_ < 1 cm^3^	NA	NA	D_max_ < 32 Gy (6.4 Gy/fx)

Abbreviations: OARs: Organs at risk; NA: not applicable, Fx: fractions; D: dose, V: volume.

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
