# Peer review of "Stereotactic Body Radiation Therapy (SBRT) for Oligorecurrent/Oligoprogressive Mediastinal and Hilar Lymph Node Metastasis: A Systematic Review"

_cancers, 2022, doi:10.3390/cancers14112680_

Round 1
Reviewer 1 Report
The authors performed a systematic review of 12 articles (out of 552) describing utilization of SBRT for oligoprogressive or oligorecurrent mediastinal or hilar lymph node. This to me stands more as a narrative review instead of systematic review. There are so much heterogeneities among the chosen retrospective studies that it is impossible to reach a fair conclusion based on these studies. I do not think this adds anything valuable to the existing literature and therefore I do not think this article should be accepted. However, if the authors choose a meta-analytic approach with Bayesian or random effect frequentist method, may be there is a potential to show something clinically meaningful out of these bunch of small retrospective studies.
Author Response
Dearest Reviewer,
We understand your reservations about the article.
The article follows the guidelines of systematic reviews. We agree on the enormous heterogeneity of the works. However, given the small number of articles we do not believe that a meta analysis can lead to definitive conclusions. On the other hand , we believe that in the light of the few data and the studies available in litterature, this topic represents a challenge for radiation oncologis.
Reviewer 2 Report
This manuscript entitled Stereotactic body radiation therapy (SBRT) for oligorecurrent/oligoprogressive mediastinal and hilar lymph node metastasis: a systemic review aimed to evaluate the feasibility and tolerability of SBRT in the treatment of mediastinal and hilar lesions with particular regard to the radiotherapy doses, dose constraints of organs at risk, and clinical outcomes. All reviewed studies were retrospective and showed extensive heterogeneity in terms of patient and treatment characteristics. Based on the reported clinical outcomes of 1-year local control, overall survival and progression free survival and toxicity profiles, this manuscript concludes that SBRT appears to be a safe and effective option in the treatment of mediastinal and hilar lymph node recurrence, with a good toxicity profile.
The major concern of this report is that based on the extracted data from these reviewed literature not possible to perform a statistical analysis to reach the conclusion.
Author Response
kind reviewer,
We partially agree with your comment. Although the studies are very heterogeneous, the efficacy and safety of SBRT in mediastinal and hilar oligometastasis/oligoprogressive have been confirmed by prospective studies such as RTOG 0813. the results of this study can be translated into this patient setting, as explained in the discussion.
Even if not confirmed by a meta analysis (impossible to perform given the small number of articles) the prognostic factors indicated in the discussion are unanimously confirmed in almost all the studies analyzed.
We understand how the robustness of the present review is not very solid for the characteristics of the studies, but we believe that this article can guide the radiation oncologist in the clinical choices when faced with a case of mediastinal or hilar oligoprogression.
Reviewer 3 Report
- English should be reviewed
- A more detailed analaysis of toxic deaths is warranted
- Low PFS rates deserve further discussion
- Comments about comments (discussion about previous discussions) do not add clarity to the manuscript
- Some conclusions are not based on the data presented
(see attached file)

Author Response
REV 3:
- English should be reviewed
The article was reviewed by a native speaker proofreader prior to submission
- A more detailed analaysis of toxic deaths is warranted
Please, See below
- Low PFS rates deserve further discussion
Please, See below
- Comments about comments (discussion about previous discussions) do not add clarity to the manuscript
Please, See below
- Some conclusions are not based on the data presented
(see attached file)
Simple summary was add.
“Present work is a review of the literature on mediastinal and hilar oligometastasis / oligoprogressive treated with SBRT. The use of mediastinal SBRT had historically been not feasible in view of the expected toxicity due to the proximity of critical structures such as the airways and esophagus. Although the heterogeneity and lack of some data in the analyzed studies, this literature review is the first published and can be a valid guide for the radiotherapist in the magament of the oligometastatic / oligoprogressive patient, with particular regard to the radiotherapy doses, dose constraints of organs at risk, and clinical outcomes ”
Line 206: Is it possible to quantify the number of patients treated after conventional RT or RT-CT, and the number of patients treated after SBRT? Is it possible to identify the number of patients truely re-irradiated on the same treatment volume?
this is an extremely interesting observation. It is a point that we talso had evaluated. However this data cannot be obtained from reading the article
Line 226:
We modify table 2 with table 3. Thanks
Tab 3: Which studies used which contraints?
We add in table 3 which studies use the constraints reported.
Line 249: A value judgement of a certain result should not be mentioned in this section
WE delete “very promising”.
Line 255: value judgment should be omitted in this section
WE delete “are encouraging”. Thanks.
Line 238: This low value call into question the concept of oligometastatic patient. This selection problem should be included in the discussion: Are these initial clinical experiences selecting properly the patients that could obtain a clinical benefit from SBRT?
Thaks. We add in discussion following sentence (Line 387)
“As regards the data relating to progression free survival, one year-PFS vary from 23% to 53.9%, while at 2 years is less than 20%. Considering that are stage IV patients, who has often come to SBRT after multiple lines of treatment, the data are encouraging. There is no doubt that this data raises a significant problem, namely the correct selection of the pa-tient. It is conceivable that patients were included who would not have benefited from the SBRT treatment from the beginning. Not only that, but in many cases purely palliative treatments have been included”.
Line 266: In order to derive useful information from the toxic deaths observed in these early series, it could be interesting to construct a table specifying: state of disease, previous treatment, previous RT dose and fractionation, toxicity type, and time to death.
Unfortunately these data cannot be obtained with precision from the studies and therefore it is not possible to construct a table with these data.
Line 320: No information regarding symptom controls has been provided in the results section. This information should be moved to the previous section.
We report the sentence also in line 266 (results)
Line 339: Author's agreement with Kowalchucks strategy (SBRT for locally advanced stage) is in contradiction with the expressed statement: "to reserve this treatment for highlyly selective patients".
Thank you for your comment: WE “change agree with the authors” with “We believe, instead, it is appropriate…”
Line 351: This reference is just a clinical case, unable to corroborate a fact.
We modified the sentence: “as decribed also in case report”
Line 353: The authors should consider if ten breast cancer patients in a specific series deserve such detailed discussion.
We decided to detail this part of the discussion to emphasize that the studies are prone to bias.
Line 364: It is commonly accepted that the high BED used in SBRT are high enough to surpass any possible cellular radioresistance. It seems that the authors consider that this is not the case for colorectal carcinoma metastases. Reference needed.
This was a mistake. We delete the sentence.
Line 370: In this paragraph the authors discuss the content of reviewed paper's discussions, but it is expected by the reader a discussion about reviewed paper's data. It could be omitted.
We have decided to keep this sentence as it is a general concept albeit derived from the discussion of a single article
Line 394: It is expected a discussion of the results, no a discussion on considerations included in reviewed papers' discussions.
WE ad: “Shahi et al. [23] found that 2 out of 3 patients (66.8%) at 1 year, and 42.9% of patients at 2 years, did not require a change in their strategy” also in results.
Line 395: In is not clear which series is referred to. These results should be included in the results section.
WE add reference [23]. This is one of the fundamental aspects of the SBRT, therefore it is a general concept and therefore, in our opinion, it deserves to be reported in the discussions, even if taken from the discussion of one of the article.
Line 427: The author's opinion about their own review is reiterative. The sentence could be omitted.
We delete “in our opinion”.
Conclusions: We rewrote the conclusion:
From the present review it is not possible except certain conclusions in consideration of the heterogeneity of the studies analyzed, and furthermore it is not conceivable to perform a meta analysis due to the limited number of studies present, However it is possible to deduce some important considerations
Ablative SBRT for oligoprogressive/oligorecurrent/oligometastatic mediastinal and hilar lymph nodes is both effective in term of Local control and safety, although the analysis is derived from small retrospective studies with relatively short follow-up, while data about PFS or OS could not be derived from these data do to the absence of control group. Safety is also confirmed by studies that have investigated SBRT for ultracentral thoracic lesions, in which the risk of fatal toxicity, albeit low, should not be underestimated. Moreover, also as a general and intuitive consideration, very close attention should be paid to the reirradiation lymph node metastasis, because there may be a high PTV overlapping with the airways with consequent risk of high toxicity. Therefore, multidisciplinary discussion is mandatory for the evaluation of the real risks and benefits on a case-by-case basis. The authors suggest that for lymph node NSCLC oligorecurrence, SBRT should be reserved for selected cases, while concurrent conventional chemoradiation is preferable, if not previously irradiated. Good performance status, small PTV volume, the absence of previous thoracic irradiation, and the administration of high BED dose seem to be factors that correlate with greater local control and better survival rate. In the presence of symptoms related to the thoracic lymph nodes, SBRT determines a rapid control that lasts over time. Moreover, also in this setting, SBRT could delay the initiation of systemic treatments.
Finally, the authors of this study recommend a RT dose delivered in 4-6 fractions with a BED greater than 80-100 Gy. Prospective studies are necessary to confirm the current evidence and to answer the questions still open.
Reviewer 4 Report
The authors presented a systematic review on “Stereotactic body radiation therapy”. Authors used the suitable and sufficient keyword to search the scientific articles to perform the review study. However, the following points should be incorporated in the manuscript. The Introduction should mention the importance to SBRT in doze dependent manner before and after the surgery. Author should also be aware of several randomized trials comparing surgery to SBRT in early-stage operable patients have unfortunately closed early due to poor accrual.
Authors should discuss the available chemotherapeutic drugs which are being used in combination with SBRT. References should be in constant format (For example, Ref 5 includes date; ref 12: journal name should be abbreviated, ref 28: title is in capitalized each word format etc.).
Author Response
REV 4:
The authors presented a systematic review on “Stereotactic body radiation therapy”. Authors used the suitable and sufficient keyword to search the scientific articles to perform the review study. However, the following points should be incorporated in the manuscript. The Introduction should mention the importance to SBRT in doze dependent manner before and after the surgery. Author should also be aware of several randomized trials comparing surgery to SBRT in early-stage operable patients have unfortunately closed early due to poor accrual.
ANSWERS:
We modified introduction including the importance of SBRT and dose depending manner, we add following sentence (introduction):
“Since its development in 1990, SBRT has emerged as one of the most significant advances in modern era, by utilizing accurate target delineation, motion management, conformal treatment planning, and daily image guidance, SBRT is able to deliver high doses in few fractions and provide a steep dose fall-off outside the target and low toxicity [1-4]. The pioneering prospective dose-finding SBRT study was conducted at Indiana University and determined maximal tolerable doses in 47 medically inoperable patients to be 3×20 Gy and 3×22 Gy for T1 and T2 lesions, respectively [5]. Subsequently, Timmerman et al. published the results of 70 patients with T1-2N0 inoperable tumors who were treated with 60 and 66 Gy in 3 fractions, despite excellent 2-year LC of 95%, toxicity was unacceptably high [6]. To date, there have been 3 randomized control trials comparing surgery vs. SBRT in operable patients (ROSEL, STARS, RTOG 1021/ACOSOG Z4099), all of which have closed due to poor accrual. Despite this, a pooled analysis of patients from the STARS and ROSEL trials offers potential insight. In this analysis, a total of 58 patients were analyzed with an estimated 3-year OS of 95% for SABR vs. 79% for lobectomy, with a median follow-up of 40.2 months for the SBRT group and 35.4 months for the surgery group [7-8]”.
Authors should discuss the available chemotherapeutic drugs which are being used in combination with SBRT.
About drugs accually used in conbination with SBRT we add in discussion following sentence (discussion):
Nowdays the interaction between SBRT and new drugs available is of great interest, infact, the advent of immune checkpoint inhibitors (ICIs) has dramatically changed the landscape of cancer care, since immunotherapeutic strategies are emerging as potentially curative systemic therapy for several tumors, especially for tumor types traditionally known to have poor outcomes. Treatment with a fully human anti- Cytotoxic T-lymphocyte–associated protein 4 (CTLA-4) antibody has been associated with long-lasting responses in several hematologic malignancies (Alatrash et al., 2016), and a high proportion of durable, complete responses in patients with advanced metastatic melanoma (Ascierto et al., 2015). Anti- Programmed cell death 1 (PD-1) or programmed death-ligand 1 (PD-L1) blocking antibodies have shown objective responses in a variety of solid tumors including melanoma, lung cancer, prostate cancer, breast cancer, ovarian cancer, head and neck cancer, and a subset of colorectal cancers.
Immune checkpoint blockers have been demonstrated to enhance durable disease responses in both early and advanced tumor settings, alone or in combined strategies, as well as improving or retaining the patient’s quality of life Some studies declare that concurrent and nonconcurrent treatment with SBRT and checkpoint inhibitors achieve better outcomes with no increased toxicity [28,29,31, 32,46,48,54,67,68]. However, others warn of possible immune-related adverse events and a synergistic effect of radiotherapy and immunotherapy on toxicities [69]. Moreover, the addition of SBRT could trigger a reactivation of the immune response in patients who are no longer responsive to immunotherapy, triggering an "ex novo" immune response.
References should be in constant format (For example, Ref 5 includes date; ref 12: journal name should be abbreviated, ref 28: title is in capitalized each word format etc.).
we have corrected and added the references to respond to the reviewers' comments.
Round 2
Reviewer 1 Report
Unless the authors use a meta-analytic approach, there is nothing much to extract from this paper.
Author Response
Thanks for your comment. as requested by the editor and the reviewer 3 we proceeded to perform a further revision of the English lanhuage
Reviewer 2 Report
No additional comments or suggestions to the revised manuscript.
Author Response
Thanks for your comment. As requested by the editor and the reviewer 3 we proceeded to perform a further revision of the English lanhuage
Reviewer 3 Report
Authors have considered the reviewer's suggestions and provided satisfactory explanation in case of disagreement.
The following minor corrections indicate that one more cycle of language checking is needed.
There is not continuity in the gap between lines 73 and 76.
Line 92: "Several experiences OF treatment OF various sites OF macroscopic lymphadenopathies" (Reiteration)
Line 388: "stage IV patients who has" (Concordance)
Line 469: "both effective in term of Local control" (in terms of local control)
Author Response
Thanks for your comment.
As requested, we proceeded to perform a further revision of the English lanhuage